# Activation Mechanism of Coal Gangue and Its Impact on the Properties of Geopolymers: A Review

**DOI:** 10.3390/polym14183861

**Published:** 2022-09-15

**Authors:** Ruicong Han, Xiaoning Guo, Junfeng Guan, Xianhua Yao, Ying Hao

**Affiliations:** 1School of Civil Engineering and Communication, North China University of Water Resources and Electric Power, Zhengzhou 450045, China; 2State Key Laboratory of Eco-Hydraulics in Northwest Arid Region, Xi’an University of Technology, Xi’an 710048, China

**Keywords:** geopolymer, coal gangue, activation mechanism, toxic curing, solid waste utilization, geopolymer property

## Abstract

Coal gangue is one of the industrial solid wastes that may harm the human body through the ecosystem for a long time. Using coal gangue in geopolymer preparation can effectively reduce cement output and meet the sustainability requirements. In this paper, the physical and chemical characteristics, including the heavy metal content, of coal gangue from different producing areas are described. Then, the mechanism of physical activation (mechanical and thermal activation), chemical activation, and compound activation of coal gangue are illustrated. The machinability, as well as the mechanical, microscopic, and toxicity consolidation properties of geopolymers prepared from coal gangue, are summarized and analyzed. The results indicate that the coal gangue geopolymers can have higher mobility and mechanical strength than cement-based composites by adjusting high calcium element material, alkali activator content, Na_2_SiO_3_ modulus, and curing condition. After physical activation, coal gangue is used in geopolymer preparation with a chemical activator (alkali excitation agent), which effectively forms a three-dimensional silicon aluminate polymer network. The pore structure is dense, the physical fixation and chemical bonding are strengthened, and the solidification and adsorption of heavy metal ions are improved. Further, it can also be applied to solidifying radioactive waste, which is following the future development direction.

## 1. Introduction

With the world’s attention on the circular economy, coal enterprises and organizations have been committed to fundamentally changing the economic growth model to achieve a circular economy. According to the BP Statistical Yearbook of World Energy 2020, global coal production decreased slightly compared to last year, and the output could reach 7742 million tons [1]. Figure 1 shows the year-on-year worldwide coal consumption growth since 2010. Coal consumption in China, North America, Europe, the USA, and India significantly grew before 2015. However, coal consumption after 2016 declined in Europe and North America, whereas in Asia it continued to increase till 2020 and was affected by the COVID-19 pandemic. Except for China, Indonesia, and Vietnam, which increased slightly, the rest of the countries have varying degrees of decline. Today, China accounts for about 50% of the global coal consumption, and its influence will be particularly significant [2].

Coal gangue (CG) is a hard rock produced during coal formation, which is formed at the same time as coal seam formation. CG may enter coal from the associated coal rocks in the form of excessive carbon growth. Its output generally accounts for about 10% of the coal output. China’s annual raw coal output is 3.5–4 billion tons, and the annual CG discharge is about 500–800 million tons [3]. As one of the 10 major industrial solid wastes in the world, CG is also the second-most-utilized resource [4], and its resource utilization has become a hot issue in countries where coal is the primary source of energy [5,6].

As shown in Figure 2a,b, CG accumulated into a mountain. All the CG accumulated together occupies land resources, pollutes the groundwater resources, and endangers the surrounding air quality. In addition, long-term exposure to the sun leads to the continuous accumulation of internal heat, leading to the spontaneous combustion of CG and the release of many harmful gases, such as sulfur dioxide (SO_2_) and carbon monoxide (CO), into the atmosphere. Some heavy metal elements are also released into groundwater that may harm the human body through the ecosystem [7]. In the last two decades, many countries have been actively trying to utilize CG in different industries to address the environmental issue of CG accumulation [8,9,10]. Comprehensive use of resources is mainly concentrated in landfills, road construction, power generation, brick manufacturing, and chemical industries [2,11,12,13]. However, its utilization rate and added value are still low [14].

Cement is the most commonly used cementing material in construction, especially in China, where the production of Portland cement accounts for about 60% of the global cement production [15]. Cement production involves huge carbon dioxide (CO_2_) emissions, accounting for 5–7% of global anthropogenic CO_2_ emissions [16]. As the global greenhouse effect exacerbates, the cement industry has developed various strategies to reduce CO_2_ emissions. Among them, replacing Portland cement clinker with geopolymers [17,18] and supplementary cementing materials (SCMs) [19,20], which are aluminosilicate minerals or solid wastes as raw materials, are very effective approaches.

CG is a type of important polymer material. It is mainly composed of aluminum silicate minerals, rich in silica and alumina. The component activity itself is extremely low. However, through a certain physical and chemical excitation to change its chemical composition and internal structure, the potential of gelation activity can be inspired [21]. With intense optimization, CG can be secondarily utilized in geopolymer preparation.

One of the keys to developing the industry is dealing with CG and making it widely used in the construction industry instead of cement. Therefore, although many researchers have undertaken much research on CG, no summative article is available in the published literature. In this paper, Figure 3 shows the flow diagram for the selected study. A large number of studies on coal gangue aggregate geopolymers are fully reviewed and analyzed to achieve the following objectives:(1)To summarize the physicochemical characteristics and heavy metal content of CG and analyze its performance trend.(2)To investigate all the activation methods and mechanisms of CG.(3)To discuss the effects of activated CG on coal gangue geopolymer (CGG).

First, the chemical composition of the polymer and the reaction mechanism were demonstrated. Then, the physical and chemical characteristics of the CG and heavy metal content were summarized. Additionally, different activation methods of CG and corresponding mechanisms were analyzed. Further, the impact of activated CG on the workability, mechanical properties, microstructure, and toxicity consolidation properties of the CGG were reviewed. Lastly, the discussion and conclusions were given to provide effective theoretical support for CGG research, which helps its wide application in construction.

## 2. Chemical Composition and Reaction Mechanism of Geopolymers

Geopolymers—cementitious materials—were first discovered and named by French scientist Professor Joseph Davidovits in the 1970s [22]. Geopolymer refers to aluminosilicate mineral polymers formed by geochemistry or geological synthesis. It is a three-dimensional network gel with amorphous and quasi-crystalline characteristics formed by the polymerization of silicon–oxygen tetrahedron and aluminosilicate tetrahedron as raw materials in the alkaline environment [23]. Many studies demonstrate its excellent properties, such as high early strength, good volume stability, chemical corrosion resistance, permeability and high-temperature resistance [24,25,26,27], and excellent mechanical properties and durability [28].

### 2.1. Chemical Composition of Geopolymers

Geopolymers are amorphous silicon–aluminate inorganic polymers consisting of a tetrahedral oligomer silicon–oxygen tetrahedron and aluminum–oxygen tetrahedron. These oligomers form inorganic polymers with a three-dimensional network structure by polymerization reaction under the activation of alkaline materials [29]. During geological polymerization, Si(IV) is only replaced by Al(IV). The empirical r of a geopolymer is Mn[(Si–O_2_)z–Al–O_2_] N·H_2_O, where M represents alkali metal cations such as potassium or sodium, n represents the condensation degree, W represents the number of water molecules, and Z represents the Si/Al ratio [23].

Since Si^4+^ is replaced by Al^3+^, the Si–O–Al–O– structure has a negative charge. The excess negative charge in aluminum atoms is moderated by the positive charge of sodium or potassium. The molecular structure of settlement network gels can be composed of three different monomer units. According to the ratio relationship between Si/Al in the reaction products, geopolymers can be divided into three types [30]: PS type (–Si–O–Al–), PSS type (–Si–O–Al–O–Si–), and PSDS type (–Si–O–Al–O–Si–O–Si–), as shown in Figure 4.

### 2.2. Geopolymer Reaction Mechanism

The geological polymerization process is initiated by the reaction of an active silicoaluminate source with an alkaline solution forming Si(OH)_4_ and Al(OH)_4_ monomers. Then, the monomer is converted into oligomer by polymerization, and the silicoaluminate gel is formed by oligomer condensation. For example, the low polymerization state of the sodium silicate tetrahedron group and the active aluminum oxide layer in metakaolinite have a combination reaction, i.e., the low polymerization degree of the tetrahedron group and the metakaolinite aluminum oxide layer have a “bonding reaction”. The low-poly siloxane tetrahedral group acts as a bond cooperation of “gelatinization” to “bond” metal particles, forming a net-like three-dimensional spatial structure. The reaction model is shown in Figure 5.

The reason for the performance difference between ordinary Portland cement and geopolymer can be attributed to the difference between C–S–H and C–A–S–H. The aluminum tetrahedron of the polymer can replace the silicon tetrahedron of the structure. The cross-linking between Dreiketten chains [31,32] offers greater rigidity and strength. The occurrence of cross-links chemically unbalances the compounds formed by the alkaline activation of the slag, which is why the presence of alkaline ions, preferably metals such as Na^+^, K^+^, is necessary to promote the balance of charges [33]. The types of binding systems formed depend on different precursor materials, i.e., systems rich in (Ca + Si) and (Si + Al) [34]. The reaction product of calcium-rich slag is aluminum-calcium silicate hydrate (C–A–S–H) gel. The silica-rich CG and metakaolin form sodium (calcium) aluminum–silicate hydrate (N(C)–A–S–H gel). Due to the combination of C–A–S–H and N(C)–A–S–H gels, the blend system shows improved performance and has higher mechanical properties and durability [35,36]. CG is mainly a system rich in (Si + Al), which has very low activity. However, its potential cementitious activity can be stimulated by changing its chemical composition and internal structure through physical and chemical activation [21]. The principle of CGG preparation is to grind CG mechanically into coal gangue powder (CGP) to produce certain activity. CGP shows high activity due to the Si–O bond fracture. The aluminum–oxygen layer easily bonds with the silicon-oxygen tetrahedral group with a low polymerization degree, realizing secondary utilization as a resource.

## 3. CG Properties and Preparation Process

### 3.1. Physical Characteristics

CG has a low carbon content, and a lot of coal powder adheres to the surface of CG, which is needle-like, loose and porous, and brown and black in color. The apparent density of CG aggregate is similar to that of natural stone, i.e., about 2600 kg/m^3^, and the bulk density is about 1200 kg/m^3^. Water absorption and crushing value (about 5% and 20%, respectively) are relatively higher than those of natural stone (about 2% and 15%, respectively) [37].

After ball milling, the CG structure becomes loose and porous. The porosity increases again after calcination. Figure 6 shows the scanning electron microscope (SEM) image of calcined CG powder. After the ball-mill grinding and calcination, CG powder particles are dense and smooth at the surface, with pores and prismatic shape. After heating, calcined CG powder becomes an irregular and porous structure, which may be caused by kaolinite hydroxylation and the phase change of the metakaolin. At the same time, the carbon content promotes irregular amorphous particles and pores [38].

### 3.2. Chemical and Mineral Composition

Based on the oxide content of CG, CG can be divided into clay gangue (SiO_2_ (40–70%), Al_2_O_3_ (15–30%)), sandstone gangue (SiO_2_ > 70%), aluminum gangue (Al_2_O_3_ > 40%), and calcareous gangue (CaO > 30%) [40]. CG is mostly composed of silicon and aluminum. Table 1 shows the chemical components of CG in different regions. It can be seen that the cumulative SiO_2_ and Al_2_O_3_ content in most of them is greater than 70%, which is considered to have a certain volcanic ash effect. Mineral components in CG include kaolinite, quartz, illite, feldspar, calcite, and muscovite, among which kaolinite, illite, and quartz have the highest content [41]. Although the contents of major components such as SiO_2_, Al_2_O_3_, and Fe_2_O_3_ in CG in south and north China are not very different, their volcanic ash activity index is significantly different. The total alkali content controlled by geographical conditions is one of the main reasons for the difference in volcanic ash activity [42]. The kaolin content in CG in the northern areas is higher than that in the southern areas, which is related to the high silicon and aluminum content of clay coal gangue (also known as coal measure kaolin). CG in the south contains more Muscovite and illite minerals, which are mica-type coal gangue [21]. As raw materials for cementing materials, CG activity in the north is higher than in the south region.

Figure 7 shows the XRD pattern of CG. The figure shows that the main minerals include quartz, kaolinite, and muscovite. Both kaolin (Al_2_(Si_2_O_5_) (OH)_4_) and quartz (SiO_2_) have sharp crystallization peaks, indicating high crystallinity, which may hinder the formation of hydration products to provide strength. However, CG has a high potential activity and can effectively assist cement in preparing cementitious materials by stimulating its activity. The microstructure of CG is shown in Figure 8. In Figure 8a, the feldspar has become sericite, and albicite twins and cross twins are visible in the feldspar. Figure 8b shows that the microstructure of one particle is enlarged, and the latter shows the overall microstructure of a large amount of quartz. In Figure 8c, the darker minerals and detrital in the silicite are disorganized in the particles of other minerals [49].

### 3.3. Heavy-Metal Element Content

The contents of heavy metals in different CG samples are shown in Table 2. According to the standard limits of hazardous ingredients stipulated in GB 5085.3-2007 (Leaching Toxicity Identification Standard for Hazardous Wastes), Pb, Cr, and Ni in CG exceed the prescribed limits. According to the results in Table 2, excessive heavy metal content of CG is the main factor restricting the secondary utilization, and heavy metal leaching should be considered in the preparation of geopolymers. Different ages of formation and weathering are the two main reasons for the differences in the heavy metal contents of CG. The results of 75 CG samples with different geological ages show that the contents of Pb, Se, and As are the highest in the samples formed in the Late Carboniferous and Early Permian. The contents of Cr, Cd, Be, Cu, and Zn are the highest in the samples formed in the late Permian. The later the formation of CG, the lower the overall enrichment degree of heavy-metal elements [58]. In addition, five types of CG with different weathering degrees were collected. The results showed that, with the increase in weathering degree, the contents of Pb and Cu first decreased and then increased, and the contents of Cr and Zn reached the lowest in the middle stage, indicating that CG may release and pollute the surrounding soil and water [59].

### 3.4. Preparation Process of Coal Gangue Powder

CG has high crystallinity and low activity, making it difficult to generate hydration products to provide strength. To use CG as a geopolymer raw material, CG needs to be ground into powder to improve its activity. Figure 9 shows the preparation process of CGP. CG is first transmitted and accumulated around the track. The rotor then crushes it. The broken chinks are fed into the ball mill for grinding and eventually sieved to obtain a size under 80 microns. The CGP has better reactivity in this form [64].

## 4. Method and Mechanism of CG Activation

CG has a relatively stable (crystalline) chemical structure and, thus, has less pozzolanic reactivity than other geopolymer raw materials. For the active component in CG, finding ways to destroy its lattice structure and increase its amorphous phase is critical, which is the most important part of using CG as a cementitious material. Many researchers have developed various methods to activate CG, which can be classified into three types—physical activation (mechanical activation, calcination activation, and microwave activation), chemical activation, and compound activation.

### 4.1. Physical Activation

#### 4.1.1. Mechanical Activation

Mechanical activation is the physical and chemical change in CG by grinding or ball milling. The lattice distortion and local destruction of the mineral under the action of mechanical force will increase its internal energy and reactivity. The mineral components in the material are more likely to precipitate. Compared with other activation processes, mechanical activation is easy to operate, has high safety, and has stable material performance [65,66,67]. Moreover, the experimental results demonstrate that mechanical grinding can reduce the particle size, increase the specific surface area, and make the particle distribution more uniform. The results prove that the amorphous phase content is further increased and the reactivity of CG particles is improved [68].

Figure 10 shows the micromorphology of CG with different mesh numbers before and after mechanical activation. In Figure 10a, CG has a compact structure, weak adsorption capacity on a solid surface, and a small and smooth contact area, which weakens the chemical reaction rate of CG. Figure 10b–f show the CGP with five mesh sizes (0–100, 100–180, 180–300, 300–400, and over 400) after mechanical grinding. With the increase in mesh sizes, the structural micropores, internal voids, and cracks of particles increase. Accordingly, the surface area of CGP is enlarged, the free energy of surface particles is increased, and the surface adsorption capacity of CGP particles is enhanced [69]. Li et al. [70] proposed that with the increase in grid number, the original structure of CG might be destroyed, and the active reaction sites increase [71], so that the potential cementitious activity would be improved.

In addition, some scholars have studied the mechanical grinding time of CG. As the grinding time increases, the particles experience the process of minimization, agglomeration, and dispersion. After grinding for 10 h, the particle size will significantly decrease, effectively improving the activity of CG, and the bulk density will reach the maximum value of 600 kg/m^3^ [38]. On this basis, studies have shown that when the particle size reaches a certain fineness, the increasing trend of CG activity decreases, which may be due to particle agglomeration during the grinding process, so it is necessary to control the grinding time to avoid energy waste [72].

Although the kaolinite structure in CG is deformed after prolonged grinding, it cannot completely change the mineral phase. Mechanical grinding is used to increase the surface area and help activate the volcanic ash activity.

#### 4.1.2. Calcined Activation

The main reaction of CG calcination activation is the dehydroxylation of clay minerals, such as kaolinite (the endothermic reaction of kaolinite) [73], forming metakaolinite, an amorphous mineral with high volcanic ash activity. Figure 11a shows the XRD patterns of CG calcined at different temperatures. It can be seen that kaolinite and quartz are the main crystalline minerals. After calcining at 550 °C, the diffraction peak of kaolinite weakens and finally disappears at 650–850 °C when kaolinite is transformed into metacolinite. When the calcination temperature is increased to 950 °C, the gentle diffraction peak of mullite appears. 

XRD was used to test CG calcined at different temperatures. It has been found that there is a low-temperature activation zone between 500 and 800 °C, and kaolinite is dehydrated to form metakaolinite. Mullite appears when calcined at 1000 °C, and it is determined that the activity is the highest when CG is calcined at 700–750 °C [74,75,76,77,78]. After CG calcination, the activity indexes were studied based on the dissolved contents of silica and alumina. When the calcination temperature was 700 °C, the dissolved contents of SiO_2_ and Al_2_O_3_ reached 92.31% and 64.44%, respectively [79].

Some researchers have also used thermogravimetric methods (TG-DTA) to analyze the calcination activation of CG, as shown in Figure 11b. It is observed that the total weight loss of CG is about 13.41% when it is heated from room temperature to 930 °C. Among them, the weight loss is 6.27% at 0–300 °C due to the removal of adsorbed water in CG and the gradual combustion and exothermic emission of residual carbon during the heating process [80]. At 535–720 °C, the weight loss is 2.02%, indicating that kaolinite gradually absorbs heat, leading to lattice destruction and dehydration to form amorphous metakaolinite. An endothermic valley appears after 900 °C, and metakaolinite may recrystallize into spinel and other minerals when the activation temperature continues to rise [70]. 

In conclusion, calcination activation is considered the most promising method used to destroy the crystal structure and increase the reactivity of CG active substances. We can describe the induced thermal induction process with the following reaction:Al_2_O_3_·2SiO_2_·2H_2_O (kaolinite) → Al_2_O_3_·2SiO_2_ (metakaolin) + 2H_2_O (at 900 °C)(1)
2(Al_2_O_3_·2SiO_2_) (metakaolin) → 2Al_2_O_3_·3SiO_2_ (silicon aluminum spinel) + SiO_2_ (at 500 °C)(2)
2Al_2_O_3_·3SiO_2_ (silicon aluminum spinel) → 2Al_2_O_3_·SiO_2_ (mullite) + 2SiO_2_ (at 1000 °C)(3)

#### 4.1.3. Microwave Activation

Microwave activation refers to microwave radiation heating, which changes the crystal and chemical structure of CG through its thermal effect. It is a fast and efficient new heating technology [81]. Compared with traditional thermal activation, it has high efficiency, low energy consumption, and no secondary pollution during treatment [82]. After microwave radiation, heat is transferred outwardly in a circular radial shape. Figure 12 shows the temperature of CGP after microwave irradiation. CGP reaches the melting state at 400–500 °C. Small sintered particles are formed when the temperature is between 500 and 600 °C. When the temperature exceeds 600 °C, about 4–5 cm of block material is observed in the center of the CGP. This is because the indirect contact surface with microwave CGP particles increases with the increase in temperature. Thus, the grain boundary area is increased and rearranged. The viscous flow of particles and plastic flow is induced along with mass transfer, leading to particles close to each other and gradually forming the grain boundary, reducing the pore volume and holes, and forming part of the hard polycrystalline sintered body [69].

The mineral composition of CGP is changed by microwave activation. Its active substances are separated, and its particles become rounder and finer, thereby increasing the volcanic ash activity. The strongest activity has been recorded in the microwave temperature range of 600–650 °C [47]. Microwave oven power level, activation time, and water bath temperature affect the activation. When microwave oven power is 595W and the activation time is 20 min, greater active substances can be obtained [83]. Further, the influence of CG activated by microwave has been simulated by finite element software ANSYS. After microwave heating for 15 min, the temperature of the CG central node could reach about 640 °C, reaching the active condition. After heating for 15 min, the temperature of the CG central node could reach about 720 °C. It has good activity conditions [84] and its results are similar to those of the studies mentioned earlier.

### 4.2. Chemical Activation

Chemical activation involves adding strong bases (e.g., Na_2_SiO_3_, NaOH, and KOH) or acidic solutions (e.g., HCL or Na_2_SO_4_) into CG to enhance its surface area and porosity by depolymerizing, removing mineral impurities, and dissolving the outer layer. Thus, the cementing properties of CG are stimulated. The strength of CGP mortar specimens with different chemical activators at the same water–binder ratio is shown in Figure 13. With the increase in CaCl_2_ concentration, the strength of the specimen first increases and then decreases. Ca^2+^ accelerated the calcium aluminum–silicate hydrate (C–A–S–H) formation. The high charge density and mobility of Cl^−^ contribute to the diffusion of alkali solution [85]. However, the reaction of excess Ca^2+^ with OH^−^ weakens the local alkalinity and precipitation of the liquid phase. With the increased concentration of Na_2_O_n_SiO_2_, the specimen strength decreases and then increases, causing enough [SiO_4_]^4−^ and dissolved Ca^2+^ to form C–S–H gel under the “concentration effect”. With the increase in Na_2_SO_4_ concentration, the strength increases, the Na_2_SO_4_ solution erodes effectively, and the Si–O and Al–O bonds on the surface of CGP decompose [86]. With the increase in Ca(OH)_2_ content, the strength of the specimens first increased and then decreased. The OH^−^ ions in the Ca(OH)_2_ solution contribute to the decomposition of Si–O and Al–O bonds. Ca^2+^ promotes the formation of calcium aluminosilicate hydrate [87]. However, when the addition of Ca(OH)_2_ exceeds 8%, it cannot diffuse in time and form crystals with weak interfaces during the hydration reaction [69]. 

Qin et al. [88] studied CG mortar test blocks prepared using three chemical activators—Ca(OH)_2_, Na_2_SO_4_, and CaCl_2_. Except for the test blocks prepared by Na_2_SO_4_, the mechanical characteristics of the test pieces prepared by other activators were similar to those mentioned above. This result may be attributed to the insufficient dispersion of Na_2_SO_4_ during hydration and the insufficient activation of CGP. Na_2_SiO_3_ with the modulus of 2.3 was used to replace NaOH as a liquid chemical activator to study the strength of CGG. The results showed that the compressive strength of all samples increased first and then decreased with the addition of Na_2_SiO_3_. When the Na_2_SiO_3_ mass ratio increased from 0% to 50%, the compressive strength of samples was the highest and increased by about 207% [89]. Several studies [90,91] have also reported that Na_2_SiO_3_ content increases the reaction intensity. The highest effect is obtained at a mass ratio of 80% Na_2_SiO_3_, which favors polymerization and produces more reaction products. This may be because Na_2_SiO_3_ can provide a large amount of active SiO_2_ to react with Ca^2+^ and Al^3+^ to generate calcium silicate hydrate (C–S–H), which promotes the further hydrolysis of CG and sodium silicate. However, the decreased NaOH in excessive solution is not conducive to the hydration reaction. In this regard, Han et al. [92] confirmed that Na_2_SiO_3_ could be hydrolyzed to NaOH and soluble silica gel, which helps improve the CGG strength.

### 4.3. Composite Activation

Compound activation uses the two approaches mentioned above to benefit from their advantages. Mechanical activation increases the contact area of particles, and the crystal structure of kaolin is destroyed, transforming it into partially ordered semi-crystalline metakaolin and increasing its amorphous phase content. Then, calcination activation can remove the carbon component in CG and generate more amorphous metakaolinite with pozzolanic properties [93]. Several studies have shown that the mechanical activation followed by calcination can more effectively destroy the Si–O–Al(VI) synthesis in kaolinite and leads to the formation of amorphous metakaolinite with a Si–O–Al(IV) structure. It has been confirmed that the traditional mechanical–thermal activation (TMTA) technology significantly increases the CG activity [38,71]. Some researchers have also investigated the extraction rate of Al from the particle size perspective. CG activated at a lower calcination temperature (400–600 °C) was used, and the results demonstrated that mechanical activation energy could reduce the particle size and increase the volume percentage of small particles, while the reduction in particle size could improve the leaching rate of Al [94].

The increased activity of CG volcanic ash contributes to the amorphous transition of minerals during thermal and chemical activation. The compound thermochemical activation method shows that the compressive strength of the mortar composed of the calcined mixture of CG and CaSO + CaO increased by 10 MPa compared with the thermal activation alone. In addition, CaO can also reduce the crystallinity of SiO_2_ during calcination [95]. Calcination and calcium addition were also used to activate CG. Calcination and activation of CG contribute more to the strength than calcination and activation alone. The activated CG system obtained at 1050 °C has the optimal strength [96].

## 5. The Performance of CGG

### 5.1. Fresh Properties

Setting time and fluidity affect the machinability of CGG. Figure 14 shows the effects of different alkali activators and blast furnace slag (BFS). The increase in BFS content, NaOH molar concentration, Na_2_SiO_3_ modulus, and SS(CaSO_4_·2H_2_O + NaOH + Na_2_SiO_3_) activator dosage can shorten the initial and final setting times of CGG [64]. Additionally, the increase in BFS content increases the number of calcium ions. The formation of C–A–S-H and N(C)–A–S–H gels is accelerated [97], and the increasing molar concentration of NaOH could effectively dissolve CGP. In addition, the increase in the modulus of Na_2_SiO_3_ and the amount of SS activator provides a large number of active SiO_3_ tetrahedra and calcium ions, effectively improving the polymerization reaction [64]. When the CGP content is adjusted from 60% to 40%, the modulus of Na_2_SiO_3_ increases from 1.2 to 1.6, resulting in the increase in CGG flowability from 181 to 252 mm [98]. The decrease in CGP can release water, while the increase in Na_2_SiO_3_ increases the concentration of silicate ions adsorbed on the particle surface. The magnitude of the repulsive double layer power is increased, thereby reducing the reduced shear stress [99]. With the increase in Na_2_SiO_3_ base activator content in CGG, electrostatic repulsion between particles leads to greater dispersion of free water on particles and easier dispersion of particles. The dynamic yield stress and thixotropic area decrease with the increase in Na_2_SiO_3_ base activator content [100].

In addition, increasing the liquid–solid ratio and adding fly ash are measures used to increase the fresh mixing performance. A higher liquid–solid ratio leads to a larger distance between particles, weakens the interaction between particles, and improves the processability of CGG slurry [101], while fly ash particles can improve the fluidity of slurry by using the morphological effect [102].

### 5.2. Mechanical Property

CGP contains a lot of silicon and aluminum and the obtained CGG has superior strength. According to the activation mechanism above, the preparation of CGG on this basis can effectively improve the chemical excitation (as shown in Table 3). In addition, CGP is rich in silica–aluminum, highly active cementitious materials but has very little calcium, hindering the formation of C–A–S–H gel. As shown in Table 3, in order to improve the calcium content of CGG, scholars added blast furnace slag (BFS), sludge (S), calcium carbide residue (CCR), red mud (RM), slaked lime (SL), and other materials rich in calcium elements. The increase in calcium content (over 30%) is expected to improve the low reactivity of CGG [103]. For example, Li et al. [89] prepared geopolymers by adding CCR and showed that the corresponding compressive strength was significantly improved compared to the pure CG geopolymer. The 3D compressive strength of samples with 10%, 20%, 30%, 40%, and 50% CCR content increased by 202.2%, 327.6%, 325.4%, 282.8%, and 221.6%, respectively. In addition, Table 3 also reflects some problems. In order to obtain the geopolymer of coal gangue with good strength, the content of coal gangue is 50–60%, and grinding or calcination is required, which greatly increases the economic cost and reduces the utilization rate of coal gangue.

Alkali exciter is an essential reagent for the preparation of geopolymers. Alkali exciter increases the surface area and porosity of CGP particles by depolymerizing, removing mineral impurities, and dissolving the outer layer. It then destroys the silica–aluminate skeleton and dissolves SiO4^4−^and AlO4^5−^, which then react with CaO and Ca(OH)_2_. C–S–H and calcium aluminate hydrate (C–A–H) are generated [49]. The commonly used base activators are NaOH, Na_2_SiO_3_, and KOH reagents. However, due to the high modulus of commercially available Na_2_SiO_3_ reagents (more than 2.56), NaOH will be added to reduce the modulus of Na_2_SiO_3_ in the preparation of CGG. The effect of NaOH and Na_2_SiO_3_ mixed reagent on the strength of CGG was studied. The results showed that the 28d compressive strength of CGG prepared with Na_2_SiO_3_ and NaOH mixed reagent was 1.73 times higher than that of NaOH reagent. The analysis showed that the optimal concentration of the mixed activator was 18–20%. The optimal modulus is 0.6–0.8 [64,104]. Guo et al. [98] proposed that the modulus of Na_2_SiO_3_ affects the polymerization reaction. When the modulus of Na_2_SiO_3_ is too high, excessive silicate ions are generated, affecting the slag hydrolysis and reducing the precipitation of calcium ions. The low modulus of Na_2_SiO_3_ leads to the reduction in the content of Si–O tetrahedra and the reduction in the CG bonding reaction. Therefore, the appropriate base initiator is conducive to destroying Si–O and Al–O bond breaking and increasing the bond reaction in polymers.

The calcined coal gangue can effectively improve the compressive strength. After adding coal gangue (500–900 °C), the 3 and 7 d compressive strength significantly increased by 7–32% and 4–27%, respectively, with the highest compressive strength at 800 °C [105]. Researchers have also studied the curing temperature effects on the resulting properties of CGG. It is shown that increased curing temperature speeds up the movement between molecules and accelerates the hydrolysis of alkaline excitation agents. The compressive strength and flexural strength are, thus, increased with the increase in curing temperature. The temperature rise accelerates the damage to the structure of the aluminum–silicon glass network in CG. However, it can lead to high CGG concrete hardening in a short time. The rapid formation of the CGG frame structure hinders the polymerization reaction. The optimal curing temperature of CGG is found at 100 °C [106].

**Table 3 polymers-14-03861-t003:** Polymer strength of CG in different literature.

Cementing Material (Particles; Calcined)	Alkali-Activator	Optimum Condition	Compressive Strength (MPa)			Refs
			3 d	7 d	28 d	
Coal gangue (<80 μm; 700 °C), Blast furnace slag	NaOH, Na_2_SiO_3_	CG:BFS = 60:40, Na_2_SiO_3_ modulu: 0.8, alkali dosage: 18%	24.5	29.5	46.4	[104]
Coal gangue (<80 μm; 700 °C), Blast furnace slag	CaSO_4_·2H_2_O, NaOH, Na_2_SiO_3_	CG:BFS = 60:40, CaSO_4_·2H_2_O dosage: 6%, Na_2_SiO_3_ modulus: 0.8, Na_2_SiO_3_: 16%	17	29.8	40.1	[64]
Coal gangue (<75 μm), Calcium carbide residue	NaOH, Na_2_SiO_3_	CG:CCR = 70:30, Na_2_SiO_3_ modulus: 2.3, NaOH:Na_2_SiO_3_ = 30:70	18.7	24.7	44	[89]
Coal gangue (<80 μm), Fly ash	NaOH, CaCO_3_	CG:FA = 50:50, NaOH:CaCO_3_ = 36.4:65.6, curing temperature: 100 °C	-	71	101	[106]
Coal gangue (<100 μm), Blast furnace slag, Fly ash	NaOH, Na_2_SiO_3_, KOH	CG:BFS:FA = 50:25:25, Na_2_SiO_3_ modulus: 2.2, NaOH:Na_2_SiO_3_:KOH = 5.1:66.4:28.5	18.8	23.3	29.4	[107]
Coal gangue (700 °C), Blast furnace slag	NaOH, Na_2_SiO_3_	CG:SFS = 50:50, Na_2_SiO_3_ modulus: 1.3, liquid-to-solid ratio: 0.36	41	86	93	[108]
Coal gangue, Blast furnace slag, Slaked lime	NaOH, Na_2_SiO_3_	CG:GBFG:SL = 55:40:5, Na_2_SiO_3_ modulus: 2.7, NaOH:Na_2_SiO_3_ = 27.7:72.3	39	-	60.5	[103]
Coal gangue (<250 μm), Sludge	NaOH, Na_2_SiO_3_	CG:S = 60:40, Na_2_SiO_3_ modulus: 1.0	-	30	39.8	[109]
Coal gangue (<75 μm), Red mud	NaOH, Na_2_SiO_3_	CG:RM = 20:80, Na_2_SiO_3_ modulus: 2.21	17	22.5	26.3	[110]
Coal gangue (<80 μm), Blast furnace slag, Fly ash	NaOH, Na_2_SiO_3_, NaO	CG:BFS:FA = 50:40:10, Na_2_SiO_3_ modulus: 1.2	5.41	-	11.5	[98]

Note: Na_2_SiO_3_ modulus: the ratio of SiO_2_ to Na_2_O.

### 5.3. The Micro Performance

The effects of CG activity on CGG properties were analyzed from a microscopic perspective. The effects of different particle sizes and calcination temperatures on CGG activity were studied. The porosity of CGG with 50 mesh (CG-50), 150 mesh (CG-150), and 200 mesh (CG-200) was lower than that with 100 mesh (CG-100) and 250 mesh (CG-250) (Figure 15a). Among them, the CG porosity of CG-50, CG-150, and CG-200 was higher than that of CG-100 and CG-250 [111]. The higher the particle porosity, the easier the leaching of Al^3+^ and Si^4+^ and the denser the CGG. The pore size distribution curves of CGG calcined at 700, 600, 200, 800, and 900 °C moved to the right successively (as shown in Figure 15b). Above 500 °C, the activity of CG starts to excite. The activity increases first and then decreases, and the dissolution amounts of Al^3+^ and Si^4+^ increase first and then decrease. The amount of CGG gel formed is positively correlated with the amount of dissolved Al^3+^ and Si^4+^ [111], which leads to the first decrease and then increase in the pores of CGG [70]. The activity of CGG was tested using SEM, and SiO_2_ and Al_2_O_3_ in the early CGG could react with the residual cement slurry in the recycled coarse aggregate, filling the micropores between CGG and the recycled coarse aggregate (Figure 16a,b). However, the hydration products at the 28-day age of CGG were mainly calcium aluminate hydrate, a small amount of calcium hydroxide, and spiculate ettringite. Due to the lack of an alkaline environment, there is a large amount of unreacted free silica in the system, which produces a CGG hardening body in the form of amorphous silicate, leading to a large number of cracks on the concrete surface (Figure 16c) [107].

In addition to the effect of CG activity on CGG, the effects of alkali activators have also been studied. The molecular structure of CGG with different Na/Al and H_2_O/Al was established using Na_2_Si_2_O_5_. Within a certain range, with the increase in Na^+^ and H_2_O, the energy of mineral polymer decreases, and the average bond length of Si–O, Al–O, H–O, and O–O shortens. Additionally, the molecular structure of CGG becomes more stable. Moreover, with the increase in Na/Al and H_2_O/Al, the bond angle distribution becomes more concentrated, and the mechanical strength increases [112]. The surface of CGG is prepared with different Na_2_Si_2_O_5_ base activator content and modulus. It has been observed by SEM that a large number of relatively dense gels with different forms of flocculants, clusters, webs, and flowers were formed [104]. This is because, with an appropriate amount of alkali initiator (dosage of 18–20%, modulus of 0.6–0.8), the silica–oxygen tetrahedra is easy to polymerize in sodium silicate rapidly forming long-chain silicate oligomers with the substrate and three-dimensional mesh silica–aluminate polymer [113].

### 5.4. Heavy Metal Solidification

Curing is an important method used to treat heavy-metal-containing waste, which is economical and effective [114]. Geopolymers, as a new curing system, can effectively cure heavy metal ions. When CG was used to replace low-calcium fly ash and metakaolin to produce CGG, the leaching toxicity of Zn^2+^, Cr^3+^, Ni^2+^, and Co^2+^ solidified bodies increased steadily with the increase in ion dose (as shown in Figure 17). The maximum leaching concentration was below 1.5 mg/L, which is far below the national standard value of leaching toxicity [115]. The leaching toxicity of heavy metals of CGG cured for 90 days is lower than that of CGG cured for 30 days. With the increase in curing age, the active substances in CGG undergo volcanic ash reaction and polymer gelation, making its three-dimensional network structure more compact. Thus, it has better physical fixation and chemical bonding to metal ions. Additionally, it is not easy to leach and has good stability [116,117].

The gel-like substances of CGG are mainly amorphous N–A–S–H (sodium aluminate-silicate hydrate) and C–S–H (calcium silicate hydrate) or aluminum-substituted C–A–S–H forms which are solidified by physical and chemical means due to their porous structure and the presence of negative aluminum tetrahedral charges in these gels [118]. The leaching concentrations of Zn^2+^, Pb^2+^, and Cd^2+^ in the solidified body supplemented with 70% LZT did not exceed the corresponding standard limits. In addition, it was found that amorphous gels and new crystalline phases containing Zn^2+^ were formed, and Pb^2+^ and Cd^2+^ may become part of the amorphous phase. In addition, studies have also shown that CGG can solidify the heavy metal ions contained in itself and effectively absorb metal ions [119]. Moreover, CGG was used to adsorb heavy metals (Pb^2+^ and Cu^2+^) from the solution, and the results showed that the gel was positively correlated with the adsorption capacity of Pb^2+^ and Cu^2+^, with the maximum adsorption capacity of Pb^2+^ and Cu^2+^ being 137.7 and 90 mg/g, respectively [120].

The principle of heavy-metal solidification is that the insoluble heavy metals are mainly encapsulated in the three-dimensional network structure of CGG. The dissolved heavy metal ions react with CGG mainly through chemical bonds or pore structures. Given the solidification principle, some researchers applied it to radioactive waste. Li et al. [121] studied the fixation effect of CGG on uranium, and the results confirmed that after curing for 28 days, the lowest static leaching concentrations of solidified uranium soil were 15.94 and 21.27 mg/kg, respectively. The mechanism of this process is that uranium is physically encapsulated and adsorbed by C–A–S–H and N–A–S–H gels, and at the same time, negatively charged [AlO_4_] tetrahedra exchange ions with uranium. In this study, Zhou et al. [122] proposed that higher base activator content and lower base activator modulus of CGG promoted the increase in 3D network structure, which could not only seal more uranium but also provide more pore structure, thus, promoting uranium adsorption.

## 6. Discussion and Gap

In summary, CG has poor physical properties and many active substances. It has high crystallinity and low activity. Therefore, CGG can only be effectively prepared after CG activation. This paper focused on the activation mode and mechanism of CG and discussed and evaluated the machinability, mechanical properties, microscopic properties, and heavy-metal-curing properties of CGG.

It was found from the summary research that the following problems need to be solved before CGG is promoted and applied in structural engineering.

(1)According to different weathering ages and weathering degrees, the content of heavy metal elements varies greatly, Pb, Cr, and Ni elements exceeding the standard to different degrees, but there are few studies focused on CGG leaching, which is not conducive to the application of coal gangue preparation geopolymers.(2)There are many studies on CGG in the existing literature, but few study durability, which has a great impact on the service life of buildings. The lack of research on durability affects the promotion of gangue geopolymers.(3)The research purpose of CGG is to replace cement. Most of the existing studies are focused on the properties of cement materials. However, there are few studies on the combination of CGG and aggregate to prepare concrete, which cannot be applied in practical engineering quickly.(4)The admixture of CGG is millimeter-level mineral waste, while the research on CGG of nanoscale materials is less common. The morphology effects, activity effects, and micro-aggregate packing effects of nanoscale materials (such as nano-silica, graphene oxide, and nano-calcium carbonate) are much higher than those of millimeter-scale materials.

## 7. Conclusions

In this paper, the properties and activation mechanism of CG were reviewed, and the performance research of CGG by various researchers from various countries was summarized and analyzed in a wide range. The study aimed to expand the resource utilization of CG. The main conclusions are as follows.

(1)The CG performance difference in different parts is very significant. CG has poor physical characteristics, and most of its chemical composition is SiO_2_, Al_2_O_3_, and Fe_2_O_3_. Kaolinite is the main mineral composition. The higher crystallinity causes difficulty in generating hydration products to promote strength. In addition, because of the different sources of CG, it has different levels of heavy metal element content in the paint.(2)The CG activation methods are physical activation (mechanical activation, calcination activation, and microwave activation), chemical activation, and composite activation. With three activation methods, the activity of CG can be effectively improved, and the gelling properties of CG can be stimulated. The composite activation, in combination with other activation methods, effectively plays respective advantages, and activation is more efficient.(3)CG is a silica–alumina material. Reducing the content of CG or increasing the liquid–solid ratio of CGG can effectively improve fluidity. Increasing the content of the CGG base activator and the modulus of Na_2_SiO_3_ can shorten the setting time and improve fluidity. Adding high-calcium material can effectively promote the formation of C–S–H gel and increase the alkali initiator content as well as the modulus of Na_2_SiO_3_ to effectively dissolve the silicon–oxygen tetrahedra, improve the bonding effect, increase the curing temperature, and accelerate the hydrolysis of the polymer using an alkali initiator. Changing these factors appropriately can effectively improve the mechanical strength of CGG. The activation reaction of CG affects the microstructure of CGG. The higher the activity of CG, the denser the structure of CGG, and the increase in Na^+^ and H_2_O in CGG decreases the energy of the mineral polymer, the average bond length of Si–O, Al–O, H–O, and O–O becomes shorter, and the molecular structure of CGG becomes more stable. With the increase in Na/Al and H_2_O/Al, the distribution of bond angles becomes more concentrated, and the mechanical strength increases. CGG prepared after CG activation forms a three-dimensional network of silica–aluminate geopolymer with a tight pore structure, strong physical fixation, and chemical bonding, which can effectively solidify and adsorb heavy metal ions and radioactive wastes.

## 8. Outlook

Finally, the future development of coal gangue geopolymer (CGG) is suggested.

(1)The heavy metal elements of CG exceed the standard and the heavy metal curing of CGG needs much research.(2)Increasing the durability research of CGG can effectively popularize and apply CGG in the engineering field.(3)It is necessary to evaluate the concrete prepared by combining CGG with aggregate. The future sustainable development direction is to use CGG instead of cement in concrete.(4)Nanomaterials (such as nano-silica, graphene oxide, and nano-calcium carbonate) have been used in the research on concrete, and the use of nanoscale materials to prepare CGG should be considered in the future.

## Figures and Tables

**Figure 1 polymers-14-03861-f001:**
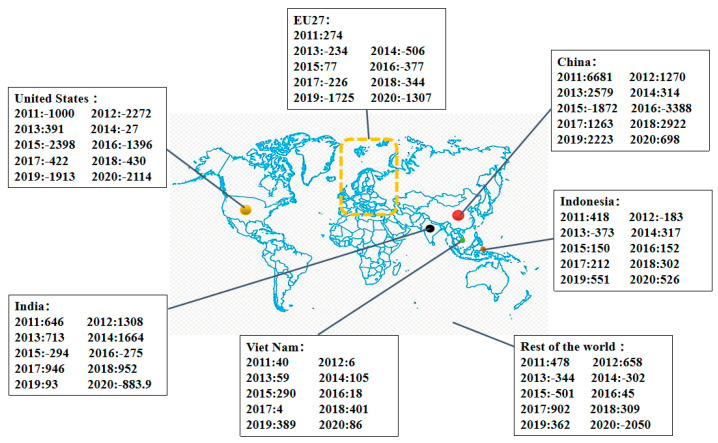
Coal consumption year−on−year growth rate, 2010−2020 (PJ) (data from International Energy Agency Statistics, 2020).

**Figure 2 polymers-14-03861-f002:**
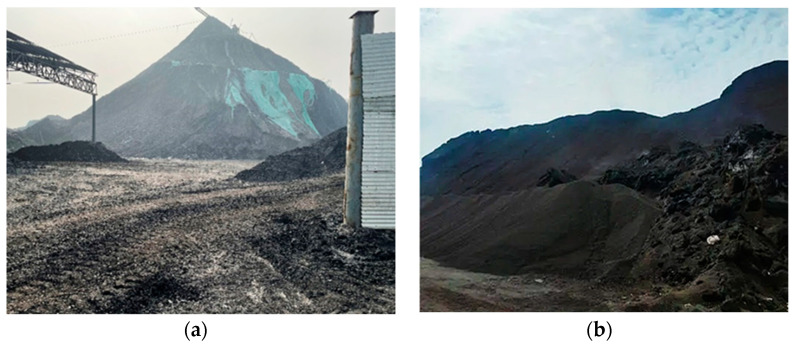
Waste coal gangue: (**a**) Coal gangue pollution; (**b**) Coal gangue mountain.

**Figure 3 polymers-14-03861-f003:**
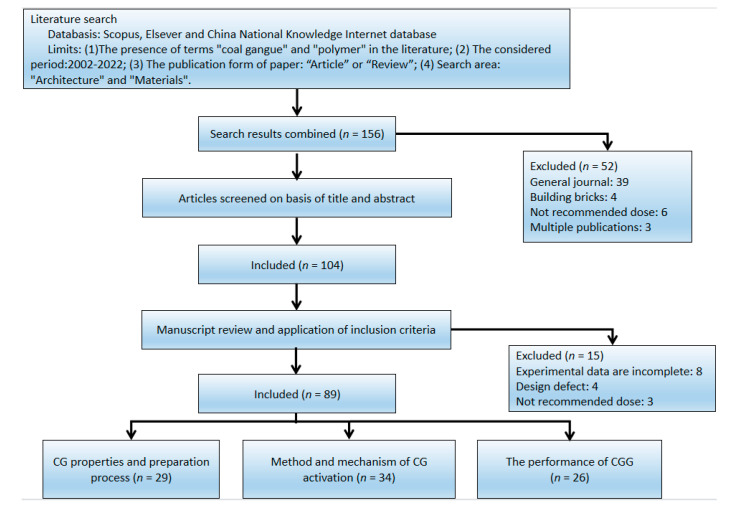
Flow diagram of the study selection process.

**Figure 4 polymers-14-03861-f004:**
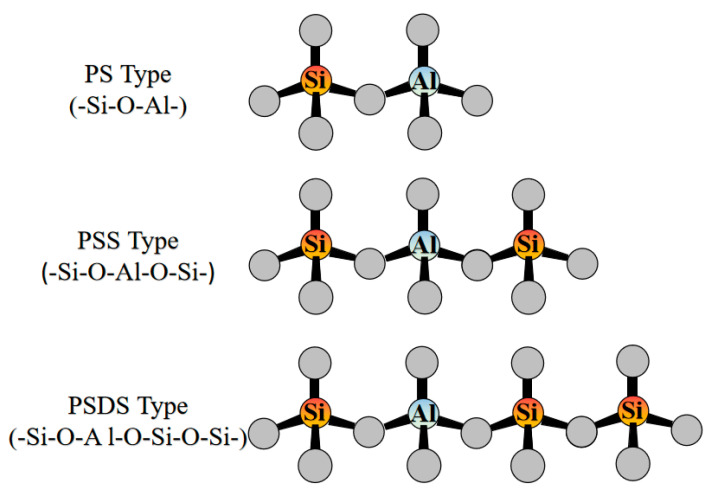
Different types of aluminosilicates [30]. (Adapted with permission from [30], Elsevier, 2022).

**Figure 5 polymers-14-03861-f005:**
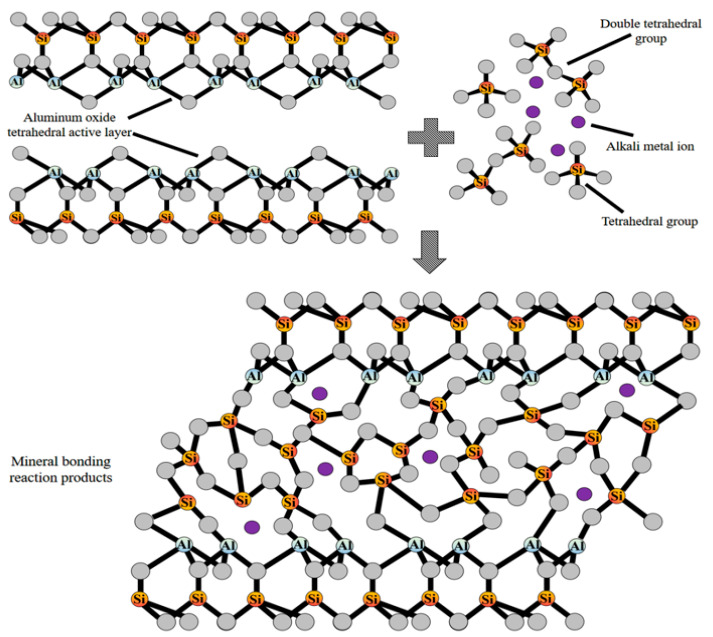
Structure model of geopolymerization.

**Figure 6 polymers-14-03861-f006:**
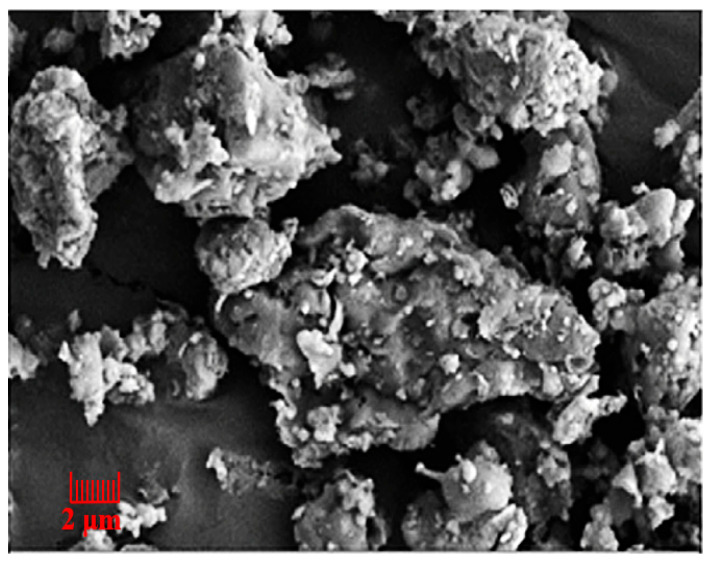
SEM of calcined coal gangue powder [39]. (Reproduced with permission from [39], Frontiers in Microbiology, 2022).

**Figure 7 polymers-14-03861-f007:**
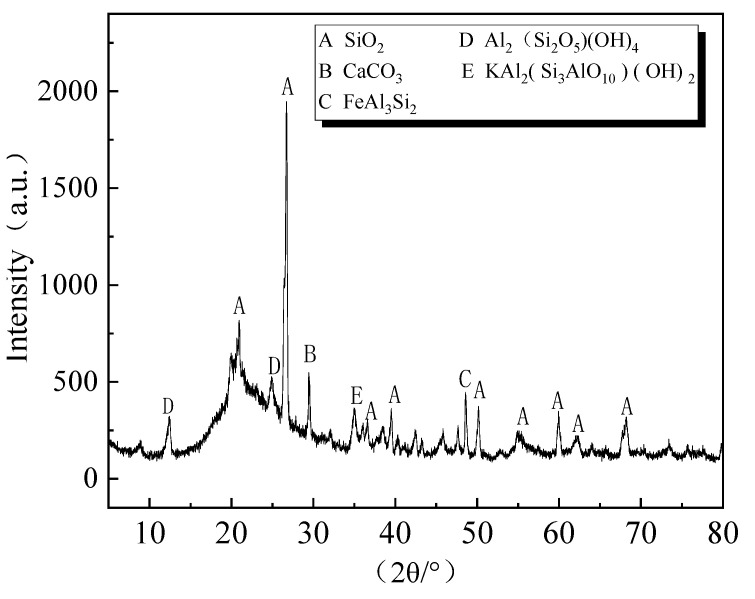
XRD patterns of CG.

**Figure 8 polymers-14-03861-f008:**
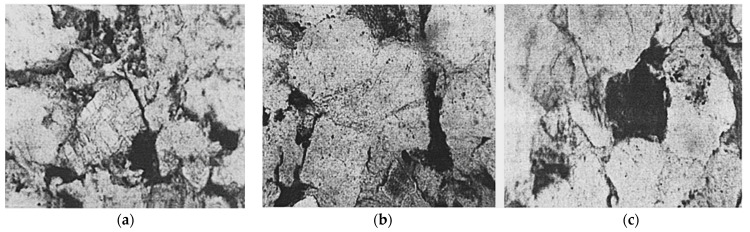
Microstructure of CG [49]: (**a**) feldspar; (**b**) quartz; (**c**) dark mineral. (Reproduced with permission from [49], Elsevier, 2006).

**Figure 9 polymers-14-03861-f009:**
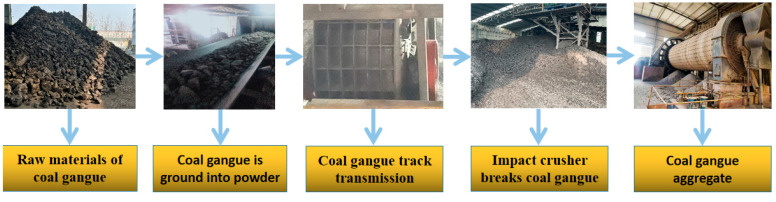
Preparation process of CGP.

**Figure 10 polymers-14-03861-f010:**
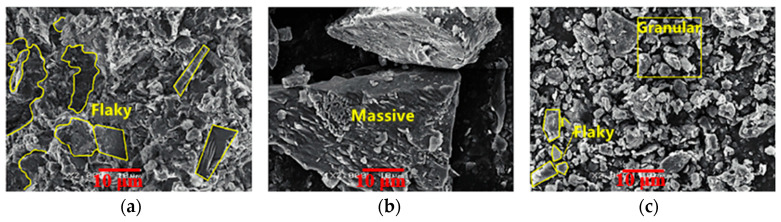
Micromorphology of the CG sample and under different meshes: [69] (**a**) CG; (**b**) 20–100 mesh; (**c**) 100–180 mesh; (**d**) 180–300 mesh; (**e**) 300–400 mesh; (**f**) Over 400 mesh. (Reproduced from [69], Hindawi, 2021).

**Figure 11 polymers-14-03861-f011:**
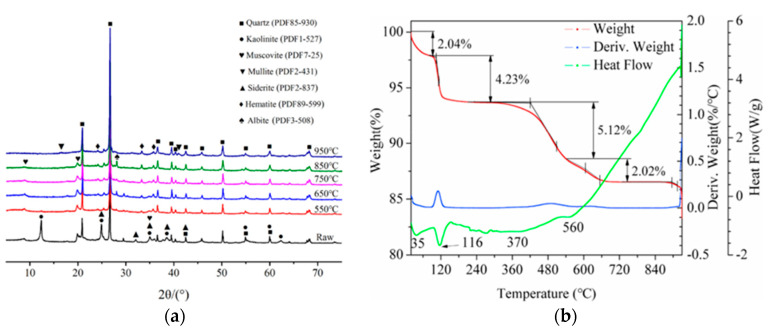
CG activity analysis at different heating temperatures: (**a**) XRD patterns of CG calcined at different temperatures [15]; (Reproduced with permission from [15], Elsevier, 2022.) (**b**) DG−TDA atlas of CG at different temperatures [70]. (Reproduced with permission from [70], Elsevier, 2021).

**Figure 12 polymers-14-03861-f012:**
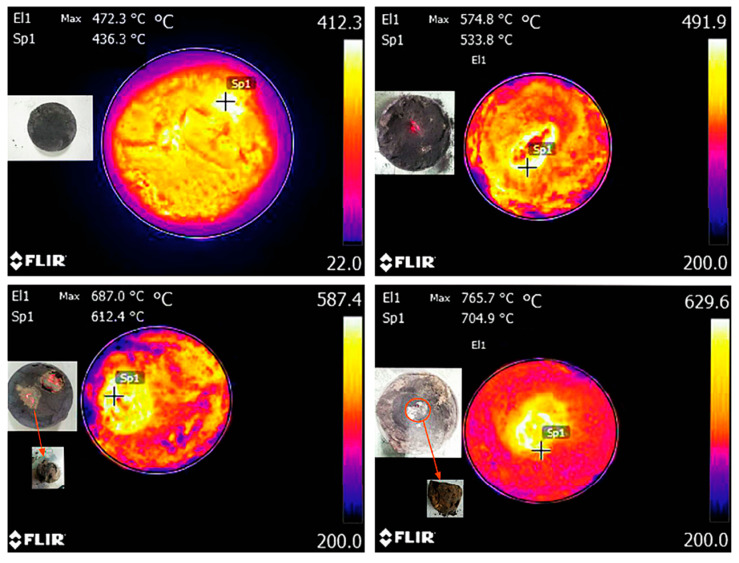
Temperature of CGP after microwave irradiation [69]. (Reproduced from [69], Hindawi, 2021).

**Figure 13 polymers-14-03861-f013:**
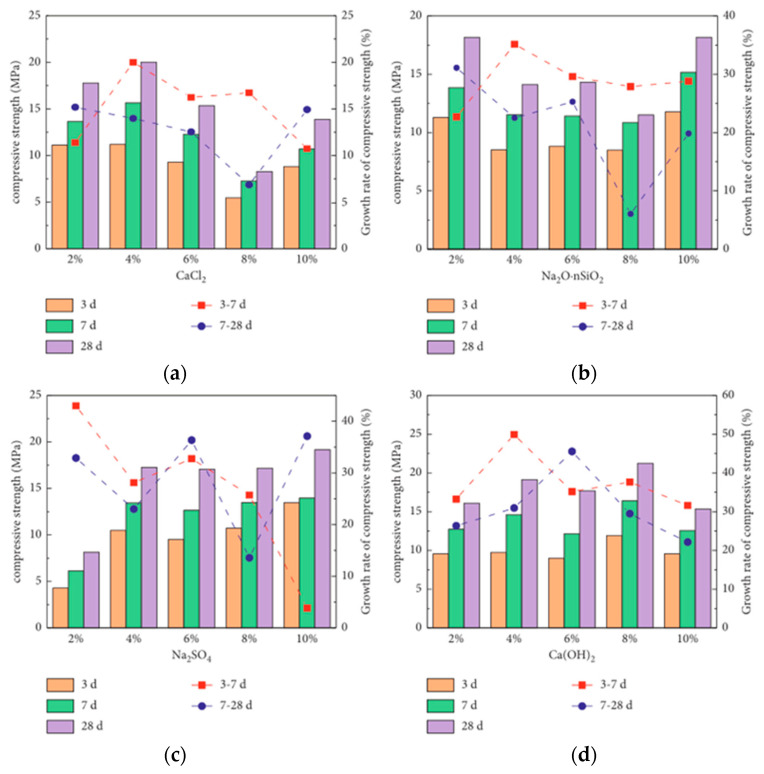
Compressive strength development of geopolymers with different chemical activators and their contents [69]: (**a**) CaCl_2_; (**b**) NaO·nSiO_2_; (**c**) Na_2_SO_4_; (**d**) Ca(OH)_2_. (Reproduced from [69], Hindawi, 2021).

**Figure 14 polymers-14-03861-f014:**
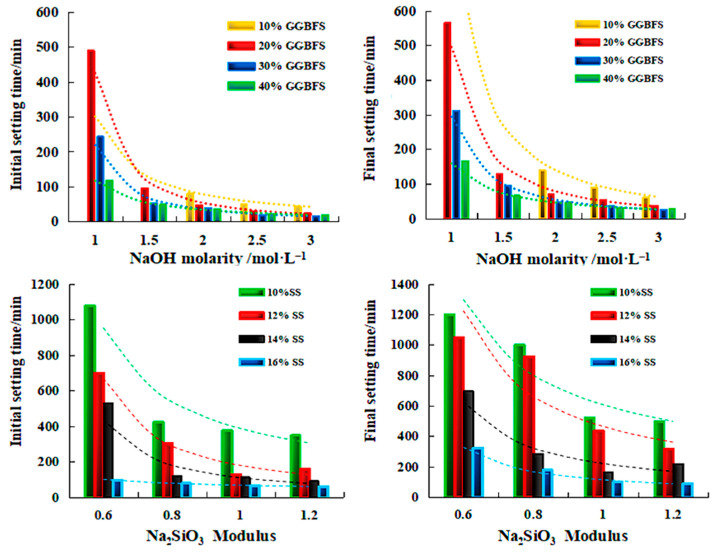
Influence of NaOH molar concentration and Na_2_SiO_3_ modulus on setting time of CGG(GGBFS: ground granulated blast furnace slag; SS: CaSO_4_·2H_2_O + NaOH + Na_2_SiO_3_) [64]. (Reproduced from [64], MDPI, 2022).

**Figure 15 polymers-14-03861-f015:**
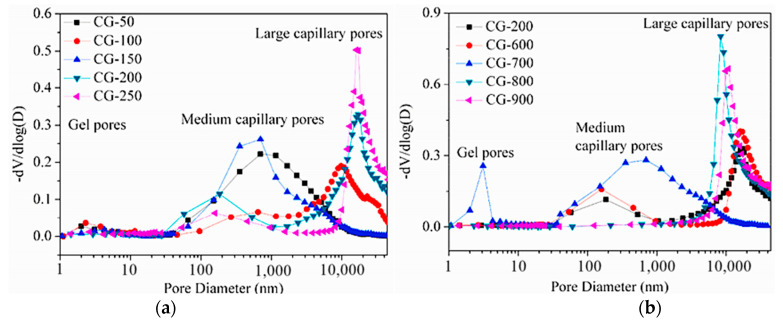
Pore size distributions of CGG at 28 days [70]: (**a**) with different particle sizes; (**b**) with different activation temperatures. (Reproduced with permission from [70], Elsevier, 2021).

**Figure 16 polymers-14-03861-f016:**
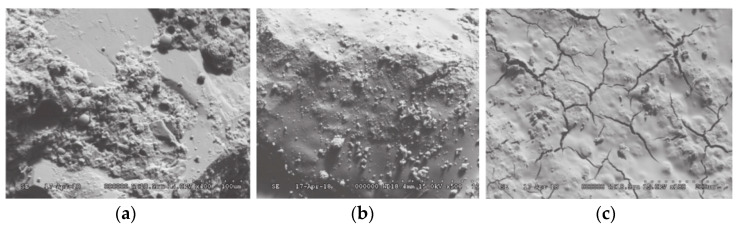
Microstructure of calcined CGG recycled concrete at different curing ages [107]: (**a**) 3 d; (**b**) 7 d; (**c**) 28 d. (Reproduced with permission from [107], Elsevier, 2019).

**Figure 17 polymers-14-03861-f017:**
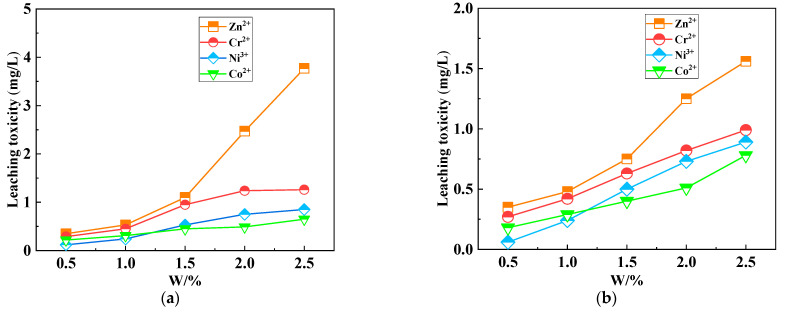
Leachate concentration contrast of different curing periods among heavy metals: (**a**) 30 d; (**b**) 90 d. (The data are from [115], Taylor & Francis, 2021).

**Table 1 polymers-14-03861-t001:** Chemical composition of CG from different producing areas.

Sources		SiO_2_	Al_2_O_3_	Fe_2_O_3_	MgO	CaO	Na_2_O	K_2_O	TiO_2_	L.O.I	Refs.
North China	Shandong	59.54	16.31	6.55	1.82	1.52	-	-	-	12.27	[43]
	Huaibei	54.12	22.38	3.56	0.71	0.98	0.64	1.13	-	-	[15]
	Fuxin	48.78	21.86	5.38	0.82	3.87	-	-	-	12.14	[44]
	Heilongjiang	58.82	27.87	8.31	-	0.78	-	1.23	1.04	1.43	[45]
	Shanxi	56.56	36.78	1.95	0.22	0.62	0.42	-	2.10	-	[46]
	Ordos	45.9	16.0	4.71	1.37	0.74	0.99	3.36	0.78	8.03	[47]
	Hebei	52.4	42.26	0.14	0.08	0.76	0.03	0.02	-	2.52	[48]
	Beijing	49.90	24.41	6.42	1.59	0.82	1.46	2.06	0.88	11.76	[49]
South China	Chongqing	58.10	24.50	5.31	1.05	5.73	1.14	1.54	-	-	[50]
	Liuzhi	46.50	16.40	13.85	3.57	10.67	1.48	1.83		-	[42]
	Yichang	49.03	34.18	0.73	-	0.20	-	0.12	1.72	13.50	[51]
	Pingxiang	52.56	16.57	3.35	2.01	1.24	0.21	2.39	-	20.71	[42]
	Xuzhou	57.95	19.02	5.32	0.82	3.16	-	-	-	-	[52]
	Xuzhou	60.24	18.50	2.58	0.52	1.48	0.14	1.53	-	-	[53]
Iran		37.8	13.14	2.85	0.73	0.76	0.28	2.02	1.17	40.96	[54]
Spain		56.4	26.34	6.42	1.07	1.06	0.17	4.02	1.21	2.38	[55]
America		34.4	14.4	3.4	0.93	1.3	0.34	3.5	1.2	38.7	[56]
Jerada		62.41	7.95	4.14	0.97	20.38	0.14	1.02	-	-	[57]

**Table 2 polymers-14-03861-t002:** Contents of heavy metals in CG (mg/kg).

Sample	Cd	As	Pb	Cr	Cu	Ni	Zn	Refs.
CG	0.1	1.56	20.19	28.72	17.17	10.59	44.54	[60]
	0.2		33.45	59.1	26.82	32.29	123.21	[61]
	0.51		20.26	46.76	60.16	16.49	78.23	[62]
	0.58	2.156	62.218	59.972	36.635		138.199	[58]
	1.097	7.71	5.087	102.84		5.131		[63]
	0.66	3.2	52.98	76.53	35.90	120.32	24.30	[59]
GB5085.3-2007	≤15	≤5	≤5	≤1	≤100	≤5	≤100	

## Data Availability

All the relevant data and models used in the study have been pro-vided in the form of figures and tables in the published article.

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
