# Peer review of "Activation Mechanism of Coal Gangue and Its Impact on the Properties of Geopolymers: A Review"

_polymers, 2022, doi:10.3390/polym14183861_

Round 1

Reviewer 1 Report

The paper "Activation mechanism of coal gang and its impact on the properties of geopolymers: A review" presents an interesting review study that has the potential to contribute to the literature on the use of waste materials in geopolymers, some specific comments are: (a) What search and paper selection criteria did the authors use to narrow down the search topic? This should be clear to the readers. (b) What real gap exists in the use of this residue in geopolymer materials? Environmental issues could be better explored in the text! (c) A deficit issue in geopolymer materials is their high economic cost, how do the authors envision this issue? (d) Fig. 2 must be named separately, in Fig. 2a and Fig. 2b, for example. (e) Authors should run new tables, similar to Table 1, with more critical discussions soon after. (f) Some main aspects of geopolymer materials should be shown by the authors more emphatically, there is a lack of some terms and applications of these materials, which can be seen in the papers: 10.1016/j.cscm.2021.e00723; 10.1016/j.cscm.2022.e01199; 10.1016/j.cscm.2022.e01332. Consider all these papers in your revised paper. (g) Authors should delve deeper into discussions about gaps in the literature on the topic!

Reviewer 2 Report

The authors of the publication rightly emphasize the variability of the chemical, phase composition of the base materials depending on the geographical occurrence, however, they focus mainly on materials originating in China, knowing that these materials change locally. Complementing the work as a review article would be adding / expanding information on the base materials from European and American countries, because despite the addition of basic information about these countries (table 1), they are quite limited, but it was not the main purpose of the work, so it is only a suggestion. there are typos in the text, such as the line a187. The presented results are very interesting and well presented, but it is necessary to add some basic information about the origin of a given CG. An example is the XRD results (line 186-191) - "Figure 6 shows the XRD pattern of CG. The figure shows that the main minerals include quartz, kaolinite, and Muscovite. Both kaolin (Al2 (Si2O5) (OH) 4) and quartz (SiO2) have sharp crystallization peaks, indicating high crystallinity, which may hinder the formation of hydration products to provide strength. However, CG has a high potential activity and can effectively assist cement in preparing cementitious materials by stimulating its activity ". The origin of the material for which the results were presented should be given, similarly to the SEM photos, despite the reference, it is good that such basic information is included in the text.

Round 2

Reviewer 1 Report

The authors have made all the suggested corrections, the paper can be accepted.